# Millimeter-Wave Conformal Directional Leaky-Wave Antenna Based on Substrate-Integrated Waveguide

Yuchen Ma [1], Xiaoya Shi [2], Junhong Wang [2],*, Yu Zhang [1], Fanqi Sun [2] and Fan Wu [3]

1   China Academy of Information and Communications Technology, Beijing 100191, China;
    mayuchen@caict.ac.cn (Y.M.); zhangyu5@caict.ac.cn (Y.Z.)
2   Key Laboratory of All Optical Network and Advanced Telecommunication Network of MOE,
    Beijing Jiaotong University, Beijing 100044, China; 18120122@bjtu.edu.cn (X.S.); 19111005@bjtu.edu.cn (F.S.)
3   State Key Laboratory of Millimeter Waves, School of Information Science and Engineering,
    Southeast University, Nanjing 211189, China; fan.wu@seu.edu.cn
*   Correspondence: wangjunh@bjtu.edu.cn

**Abstract:** Conformal antennas have been widely used in many fields due to their advantages of low air resistance and better visual appearance. In this paper, an arced conformal leaky-wave antenna (LWA) for a designable directional beam is proposed. The antenna is achieved based on a substrate-integrated waveguide (SIW). On the upper surface, a series of non-uniform transverse slots are etched. In order to guide the design of the antenna, as another key contribution of this work, a theoretical model for the traveling-wave structure is established. Using the model, the radiation property of the LWA is analyzed. In addition, by inputting the desired beam direction, the structural parameters of the LWA can be generated through the model. To verify the performance of the antenna and the model, an LWA prototype working at 28 GHz was fabricated and tested in a microwave anechoic chamber. The experimental results are in good agreement with the simulation results. The antenna achieved a gain of 9.96 dBi with cambered surface area of 1.89 $\lambda_0{}^2$. The proposed method may be a promising candidate for conformal wireless communication applications.

**Keywords:** conformal leaky-wave antenna; theoretical model; directional beam; substrate-integrated waveguide





## 1. Introduction

Conformal antennas have drawn people's increasing attention in recent years because of their wide range of applications [1–4], including mobile carriers, wearable kits, fundamental wireless coverage devices, etc. The use of conformal antennas has the advantages of reducing wind resistance and being hidden for better visual appearance. In order to guarantee and improve the dynamic range of the wireless link, research on directional conformal antennas is necessary and crucial.

To achieve directional radiation though a conformal antenna, an array method should be used. By utilizing the joint effect of all elements in the conformally distributed array, a directional beam is synthesized in the far field.

In general, the conformal antenna design is divided into two categories, i.e., active and passive methods. For the active method, beam synthesizing is achieved by active phased arrays. Among them, active phase shifters are employed to provide different phase distribution of all elements. In [5], a conformal cylindrical phased array antenna excited with composite right/left-handed (CRLH) phase shifters for switchable directional beam is proposed. In [6], a Ku-band dual-polarized cylindrical dipole phased antenna array achieving a scannable beam is put forward. In [7], a 3D-printed tightly coupled antenna array fed by seven independent ports is investigated. In [8], a wing-conformal planar inverted F-shaped antenna (PIFA) array skin for a changeable directional beam is presented.

In [9], a wing-conformal linear phased array with four-branch dipole elements fed by separate Marchand baluns is proposed.

For the passive method, conformal antennas are realized by passive feeding structures. In [10], a ground-shared and radiator-shared four-element multi-port conformal patch antenna array for wide angular coverage is designed. In [11], a Ka-band four-element standing wave conformal slot array fed by Rotman lenses is proposed. In [12], a single-feed multi-beam conformal antenna array using independently manipulated ultrathin Huygens elements is proposed. In [13], a 3D-printed conformal waveguide stand-wave slot array antenna providing a cosecant squared pattern is designed. In [14], an arced conformal array composed of four individual patch antennas achieving a beam steering antenna is realized. In [15], a wearable conformal antenna array with four broadband circularly polarized (BCP) all-textile antennas is designed. In [16], a 3D-printed conformal slotted waveguide antenna array beamformed for a normal directional beam fed by an active network is put forward. In [17], a focusing beam in near field is generated by a curved SIW, which utilizes the spatial placement of the slot elements to control the desired amplitude and phase of the excitation. In [18], a conical conformal array integrated by four double rhombic dipoles achieving dual polarization end-fire beams is developed.

Moreover, there has been some research into beam synthesizing methods on conformal antennas. In [19], a beam-steering conformal linear patch antenna array designed by deep reinforcement learning is presented. By combining optimizing excitation phases and rotating antenna elements, the fixed beam can be synthesized by a cylindrical conformal array, as described in [20]. In [21], a differential evolution algorithm is used to optimize and synthesize a sparse conformal array.

From the works above, it can be seen that most conformal antennas are designed based on standing-wave structures. For generating a directional beam, each element in some arrays needs to be fed by individual excitation or some other arrays need to be fed by a phased beamforming network.

A leaky-wave antenna (LWA) is a kind of traveling wave radiation structure with the advantage of feeding convenience. In [22], a circular LWA radiating a designable conical beam or broadside beam is proposed. Based on the structure, further research on the theoretical model for providing circularly polarized conical and broadside beam structures and a changeable beam structure have been developed [23,24]. The radiating elements on the upper surfaces of the antennas are handily fed by traveling wave structures. For conformal LWAs, comparative studies on single tapered slot radiation for far-field coverage [25] and longitudinal discrete slot radiation for near-field focused beams [17] have been investigated.

In this paper, a mm-wave arced conformal substrate-integrated waveguide (SIW) LWA with non-uniform transverse slots for a designable directional beam is proposed. To guide the antenna design which conveniently radiates to a given direction, a common theoretical model based on an arced leaky-wave antenna array established. Through the model, structural parameters of the LWA with a designable beam angle can be produced. The established model is suitable for all arced leaky-wave antenna arrays with discrete elements. The theoretical model of the conformal LWA is demonstrated in Section 2 of this paper. Using the model, the radiation-property analysis of the LWA is carried out in Section 3, the antenna design is described in Section 4 and validated by experimental results in Section 5, and finally the conclusion is given in Section 6.

## 2. Theoretical Modeling

In this section, a theoretical model is established to describe the radiation property of the conformal LWA. The radiation pattern of the conformal LWA is calculated by inputting parameters such as frequency, period quantity in transmission line, array radius, arc length of array, and so on. The key function of the model is to determine all elements' positions that make the space wave superimpose together in the expected direction in the far field to produce the directional beam. Since the structural parameters can be determined by

using this model, the antenna design is described and simplified. Moreover, the radiation properties of the antenna are easy to analyze with the help of the model.

Figure 1 describes the theoretical model of an arced array. $O$ is the origin of the coordinates where the center of the arced array is located. In Figure 1a, $R$ is the radius of the arc and $\Phi$ denotes the total angle of the arc. For a model of a leaky-wave antenna array, the traveling wave is assumed to enter from the left, as the arrow shown in the figure. Along with the arc, a series of elements is arranged. Due to the traveling wave property, elements at different positions have different phases. For different expected synthesized beam directions, elements have different distributions. The beam angle $\varphi_d$ is defined as the angle between the beam vector r and the $y$ axis. $R_e(n)$ represents the vector of the nth element parallel to $r$, starting from the nth element. $P_{slot}(n)$ is the angular position of the nth element.

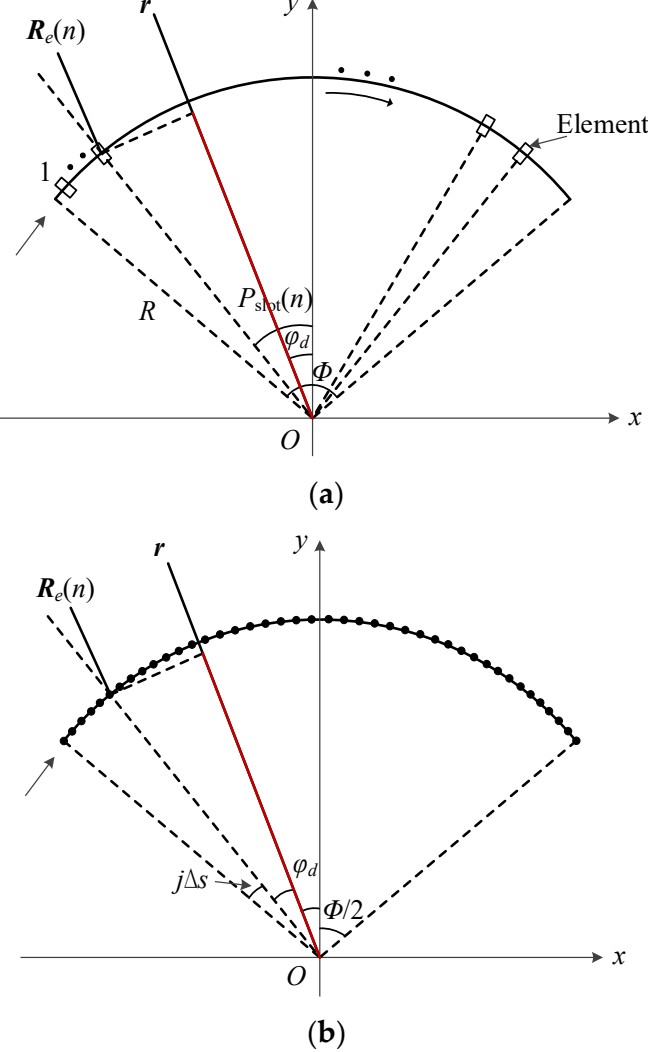

**Figure 1.** Model of an arced array (**a**) slot array, (**b**) searching for slots.

To search for all the elements that reach the in-phase condition, the first element in the arced array is assigned as the reference element. Then, the wave-path differences between the other elements and the reference element are calculated, expressed as:

$$\Delta R'_e(\varphi_d, n) = R \cos(\varphi_d - P_{slot}(n)) \tag{1}$$

For beam synthesizing, the wave phase of each element in the far field is composed of the phase of wave-path difference $\Delta\psi_{R'}(\varphi_d, n)$, the phase from traveling wave $\psi_{0'}(n)$, and the phase of the element radiation pattern $\psi_e(\varphi_n, n)$. $\Delta\psi_{R'}(\varphi_d, n)$ is given by:

$$\Delta\psi'_R(\varphi_d, n) = k_0 \Delta R'_e(\varphi_d, n) \tag{2}$$

where $k_0 = 2\pi/\lambda_0$ is the wave number in free space, $\lambda_0$ is the wavelength in free space. $\psi_{0'}(n)$ is expressed by:

$$\psi'_0(n) = -\frac{2\pi}{\lambda_g} R(P_{\text{slot}}(n) - P_{\text{slot}}(1)) \tag{3}$$

where $\lambda_g$ is guided wavelength, which is determined by $\lambda_g = L/N$, therein $N$ denotes the period quantity in the transmission line. $\psi_e(\varphi, n)$ is given by:

$$\psi'_e(\varphi_n, n) = \psi_s(\pi + \varphi_d - P_{\text{slot}}(n)) \tag{4}$$

so the total phase of each element is written as:

$$\psi_{total}'(\varphi_d, n) = \Delta\psi'_R(\varphi_d, n) + \psi'_0(n) + \psi'_e(\varphi_n, n) \tag{5}$$

When searching for the in-phase elements, the arc is discretized by a step angle $\Delta_s$, as depicted in Figure 1b. Therefore, the Equations (1)–(5) are rewritten as:

$$\Delta R_e(\varphi_d, j) = R\cos\left((j-1)\Delta s - \frac{\Phi}{2} - \varphi_d\right) \tag{6}$$

$$\Delta\psi_R(\varphi_d, j) = k_0 \Delta R_e(\varphi_d, j) \tag{7}$$

$$\psi_0(j) = -\frac{2\pi}{\lambda_g} R(j-1)\Delta s \tag{8}$$

$$\psi_e(\varphi_d, j) = \psi_s(\pi + \varphi_d + \Phi/2 - (j-1)\Delta s) \tag{9}$$

$$\psi_{total}(\varphi_d, j) = \Delta\psi_R(\varphi_d, j) + \psi_0(j) + \psi_e(\varphi_j, j) \tag{10}$$

Here, a phase tolerance $\delta$ is introduced. When $\psi_{total}$ of one element meets the in-phase condition $\psi_{total} < \delta$, the element is regarded as an in-phase element and its position is recorded in the element set. For an LWA, energy gradually radiates out of the transmission structure. Since the model is set up for a non-uniform LWA, an approximate attenuation constant $\alpha$ is considered, which is expressed as:

$$\alpha = -\ln\left(\frac{|S_{21}|}{\sqrt{1-|S_{11}|^2}}\right)/L \tag{11}$$

where $L$ refers to the arc length of the array. In this way, the amplitude of each element is written as:

$$l_\alpha(n) = 10^{\alpha l_n/20} \tag{12}$$

where $l_n$ refers to the arc length from the beginning of the array to the $n$th element. Finally, the total far field synthesized by all elements can be calculated by:

$$E_{total} = \sum E_e \times l_\alpha(n) \times e^{i\psi_{totol}} \tag{13}$$

For clarification, the schematic model diagram is summarized below, as illustrated in Figure 2.

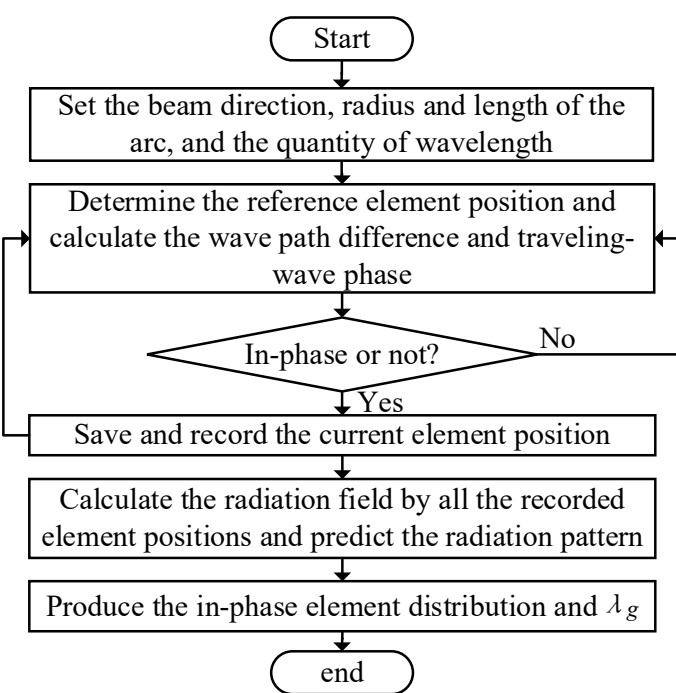

**Figure 2.** Schematic model diagram of arced array producing directional beam.

### 3. Radiation Property Analysis

In this section, the radiation property of the arced LWA is analyzed. The effects of adjusting different parameters such as $R$, $\alpha$, and $\varphi_d$ are studied respectively. In addition, the relationship between element quantity and period quantity in the transmission line and element intervals along the arc transmission line are investigated.

Figure 3a illustrates the radiation patterns of arced structures with different curvature radii based on the radiation field calculation model of the proposed arced array when setting the same beam direction, $\varphi_d$. It can be found that the expected directional beam is achieved by using the established theoretical model. Furthermore, it is possible to generate the same beam-direction pattern for arced SIW LWAs with different curvature radii by changing the position of the elements. Consequently, for cylindrical objects with different radii, the unit position distribution of the directional arced LWA can be easily generated through the model to meet the requirement. Equation (14) gives the relationship between radiation efficiency and $S$ parameters. Assuming $|S_{11}|$ is 0, which means that there is no reflection in the input port, for a given radiation efficiency the corresponding $|S_{21}|$ can be obtained, and the corresponding attenuation constant $\alpha$ can be further calculated according to Equation (11), i.e., the attenuation constant can be changed by setting different radiation efficiencies. Figure 3b depicts the radiation pattern calculated with different radiation efficiencies or different attenuation constants, in which it is shown that as the radiation efficiency increases, the calculated main beam of the directional pattern becomes narrower and the directionality becomes more prominent. Figure 4 provides the radiation patterns of different beam directions generated by the same arc structure, which indicates that by setting different beam directions in the model, the corresponding unit position distribution can be calculated. In other words, for an arced SIW LWA with the same array curvature radius and transmission line structure, different directional beams can be obtained by changing only the position of the slots.

$$\eta = 1 - |S_{11}|_2 - |S_{21}|^2 \tag{14}$$

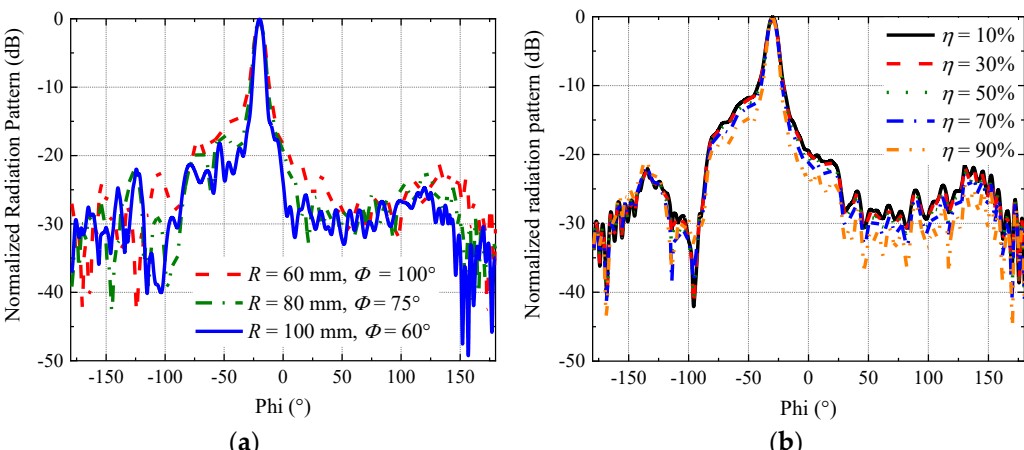

**Figure 3.** Radiation patterns when setting the same beam direction, (**a**) with different *R*, (**b**) with different $\eta$.

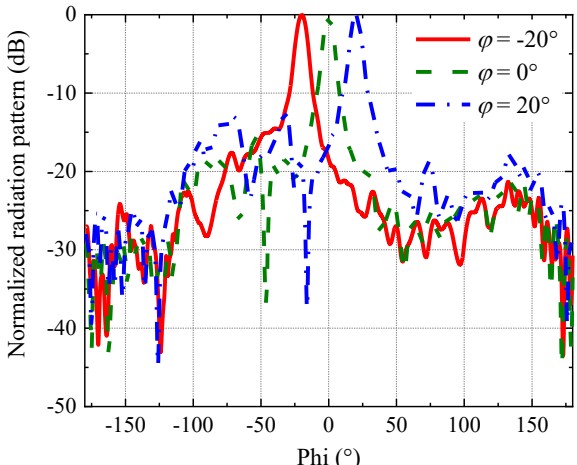

**Figure 4.** Radiation patterns calculated by different $\varphi$.

The number of elements calculated by setting a different period quantity *N* is given in Figure 5. The larger the *N* set, the more in-phase elements are found through the model.

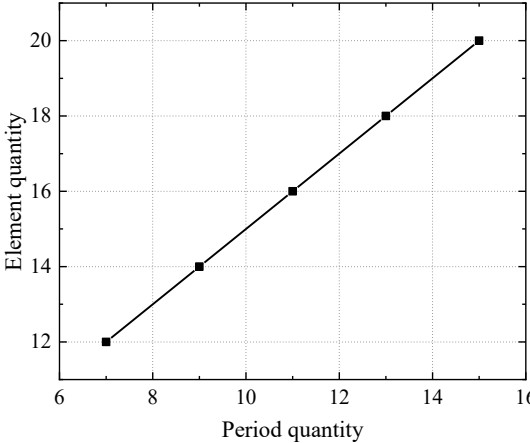

**Figure 5.** Calculated element numbers with different *N*.

Figure 6 shows the relationship between the positions of the reference elements and the number of elements obtained by the model within the first waveguide wavelength range near the feeding end. It is obvious that the positions of the reference element can be

appropriately designed on the same transmission-line structure to achieve more elements and hence radiate more energy into space.

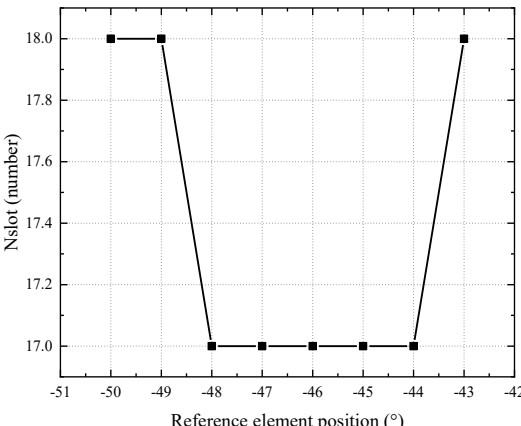

**Figure 6.** Relationship between the numbers of elements and the reference locations within the first waveguide wavelength.

The variation trend of element spacings can be analyzed from the element position distributions within the model. Figure 7 provides the variation trend of adjacent element spacing calculated on the basis of different parameters, namely beam direction, radius, array radian, and period quantity. It is clear that when the beam direction is 20° and 60°, the element spacing of the two LWAs gradually decreases along the array and tends to be flat in the rear part.

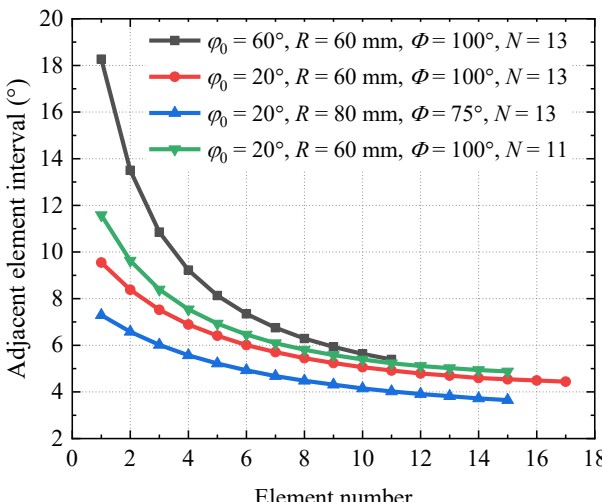

**Figure 7.** Calculated variation trend of intervals of adjacent elements with different $\varphi_0$, $R$, $\Phi$, and $N$.

## 4. Antenna Design

In the former sections, the proposed theoretical model for the arced LWA array is described, and the radiation property of the arced LWA is studied through parameter analysis based on the model. In this section, the antenna design procedure and the specific steps including expected parameters input, in-phase elements searching, radiation pattern prediction, etc., and the structural design method of the arced LWA are introduced. The array was implemented based on an arced SIW using Rogers RO3003. For directional radiation, non-uniform rectangle slots were etched on the upper metallic layer of the PCB board.

### 4.1. Design Procedure

By virtue of the established model, the arced LWA can be regularly and conveniently realized. The parameters of the LWA are provided by the model used for directivity design. The specific design procedure is presented in the following:

(1)  Declare $\varphi_d$, $R$, $\Phi$, $f_0$ and input them into the model.
(2)  Based on the model, calculate and determine the position of the reference element (the first element).
(3)  According to a given $\Delta s$, search for the in-phase element distributions in which the elements meet the in-phase condition corresponding to different reference elements.
(4)  Extract the proper set of the distribution with enough element quantity and record parameters like $\lambda_g$ from the model.
(5)  Implement structural design of the arced LWA according to the parameters produced from the model.

### 4.2. Structural Design and Analysis

Firstly, the propagation property of an arced SIW was analyzed. In [26], both the SIW and the equivalent rectangular waveguide filled with the same dielectric use the TE$_{10}$ mode, and the equivalent waveguide width of a straight SIW is expressed as [26]:

$$w_{eff} = w - 1.08\frac{d2}{s} + 0.1\frac{d2}{w} \tag{15}$$

where the diameter d and spacing s of metallic vias and the waveguide width w should meet the following criteria [26]:

$$\begin{aligned}1 < s/d < 2\\ d/w < 0.2\end{aligned} \tag{16}$$

Due to the fact that the conformal SIW is a special form of SIW, the propagation property of straight SIW was analyzed and compared with that of arced SIW. Figure 8 describes the structure of an arced conformal SIW with a metallic layer at the bottom, a dielectric layer in the middle, and a metallic layer at the top. The metallic vias are evenly spaced on both sides of the waveguide. The width between the two rows of the metallic vias, i.e., the width of the SIW, is $w$, the diameter of the metallic vias is $d$, the arc length spacing of the metallic vias is $s$, and the arc interval $s_1 = s/R$, where $R$ is the radius of the conformal SIW. Detailed dimensions of the straight and arced SIWs are listed in Table 1.

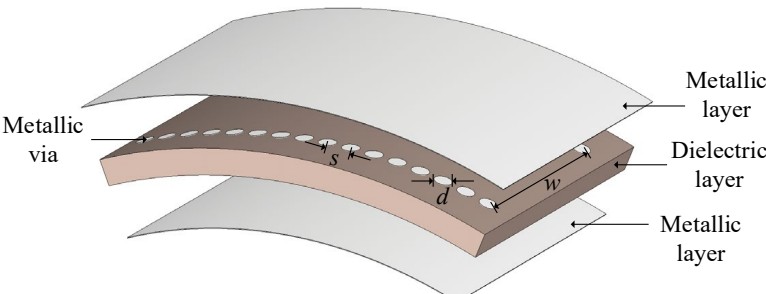

**Figure 8.** An arced conformal SIW.

**Table 1.** Dimensions of the straight and arced SIW.

| Parameters | *L* (mm) | *R* (mm) | *α* (°) | *w* (mm) | *h* (mm) | $d_{via}$ (mm) | $s_{via}$ (mm) |
|---|---|---|---|---|---|---|---|
| Straight | 55.9 | - | - | 3.88 | 1.524 | 0.6 | 1 |
| Arced | - | 30 | 104 | 3.88 | 1.524 | 0.6 | 1 |

The operating frequency was set to 28 GHz. Table 2 and Figure 9 respectively provide the cutoff frequencies and instantaneous electric field distribution of the two kinds of SIW with the same structural parameters, among which the theoretical cutoff frequencies in Table 1 was calculated by Equation (17) in [26]. It can be seen that the propagation properties of the arced SIW and the straight SIW are basically the same. Therefore, the arced SIW can be designed in line with the design method of the straight SIW [26].

$$f_c = \frac{c}{2w\sqrt{\varepsilon_r}} \tag{17}$$

**Table 2.** Cutoff frequencies of the straight and arced SIW with the same structural parameters.

|  | Straight SIW | Arced SIW |
|---|---|---|
| Theoretical cutoff frequency | 26.403 GHz | - |
| Simulated cutoff frequency | 26.385 GHz | 26.384 GHz |

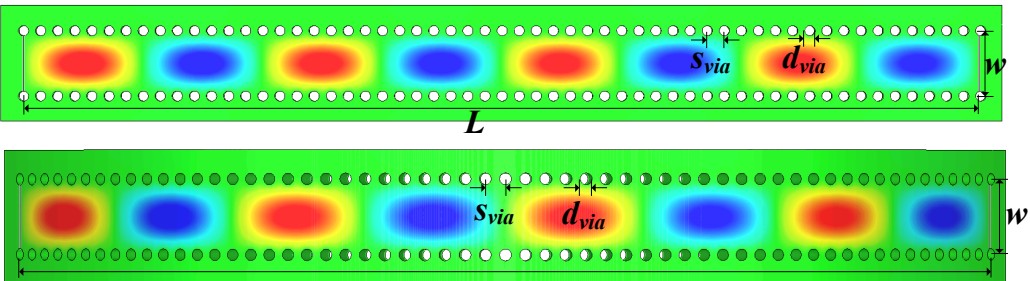

**Figure 9.** Front view of electric distributions of the two SIWs, the **upper**: straight, the **lower**: arced.

Secondly, we applied the proposed model to calculate the radiation element positions corresponding to the beam with specific direction to determine the slot positions of the arced LWA. When $\varphi_d = 20°$, $N = 13$, $R = 60$ mm, $\Phi = 100°$, and $f_0 = 28$ GHz, the element positions calculated from the theoretical model are shown in Table 2 and the structural diagram of the arced SIW LWA were established accordingly, as shown in Figure 10. The element position $\varphi_m$ in Table 3 represents the angle between the $m$th element and the first input port, and $d_m$ in Figure 10 represents the angle between the $m$th and the $(m - 1)$th element, for which $d_1$ is the angle between the first element and the input port. Taking the flexibility of the structure into consideration, the substrate material adopts Rogers RT5880: $\varepsilon_r = 2.2$, $\tan\delta = 0.0009$, the thickness of the dielectric $h = 0.254$ mm, and the two rows of metallic vias and the waveguide width comply with Equation 16. When $R$, $\Phi$, and $N$ are determined, the dielectric constant of the equivalent waveguide can be obtained by Equation (18), and accordingly the equivalent width $w_{eff}$ can be calculated by Equation (19). Then, the width of the SIW can be determined by Equation (15) [26].

$$\varepsilon_g = (\frac{\beta}{k_0})^2 = (\frac{\lambda_0}{\lambda_g})^2 = (\frac{N\lambda_0}{L})^2 \tag{18}$$

$$w_{eff} = \frac{\lambda_0}{2\sqrt{\varepsilon_r - \varepsilon_g}} \tag{19}$$

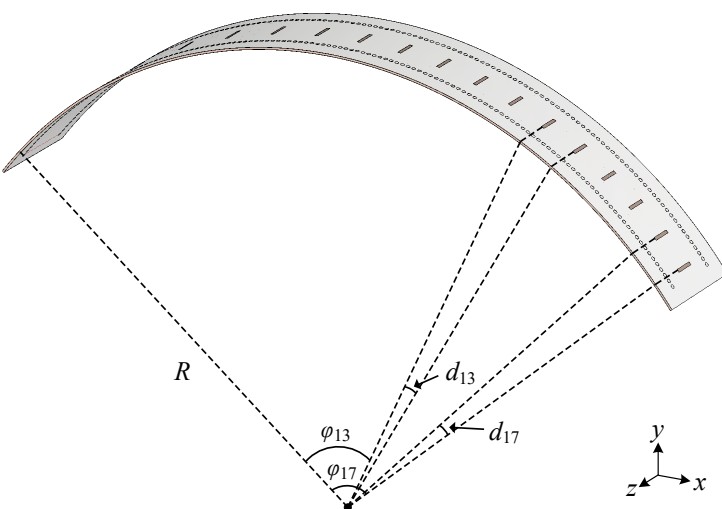

**Figure 10.** Structure of the arced antenna with non-uniform slot spacing.

**Table 3.** Element positions on the upper surface of the LWA.

|  | Element Position (°) |  | Element Position (°) |  | Element Position (°) |
|---|---|---|---|---|---|
| $\varphi_1$ | 9.54 | $\varphi_7$ | 50.46 | $\varphi_{13}$ | 80.62 |
| $\varphi_2$ | 17.92 | $\varphi_8$ | 55.92 | $\varphi_{14}$ | 85.22 |
| $\varphi_3$ | 25.46 | $\varphi_9$ | 61.16 | $\varphi_{15}$ | 89.76 |
| $\varphi_4$ | 32.34 | $\varphi_{10}$ | 66.22 | $\varphi_{16}$ | 94.26 |
| $\varphi_5$ | 38.74 | $\varphi_{11}$ | 71.14 | $\varphi_{17}$ | 98.7 |
| $\varphi_6$ | 44.76 | $\varphi_{12}$ | 75.92 |  |  |

In order to achieve good impedance matching, we considered gradually changing the lengths of the first few slots while keeping other parameters unchanged to reduce the electromagnetic wave reflection. The number of the slots with gradient lengths is set to $N_g$, and the final stable slot length $l_s$ is 2.625 mm. The slot length of the $m$th increased slot $l(m)$ can be calculated by $l(m) = l_s \times m/(N_g + 1)$. The impedance and radiation characteristics with $N_g$ set to 4, 7, and 10, respectively, were analyzed in this work. Taking $N_g = 7$ as an example, Figure 11 depicts the expanded planar graph of the arced antenna.

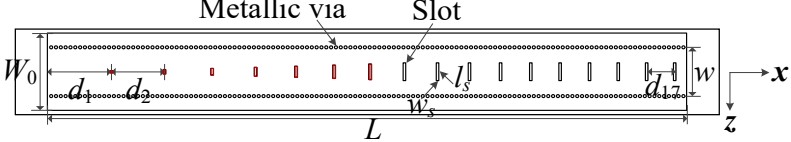

**Figure 11.** Planar graph of the arced antenna with gradually increasing slot lengths.

Figure 12 presents the radiation patterns of different values of $N_g$ and shows that the beam direction is not affected by the number of $N_g$ while the right sidelobe level rises with the increase of $N_g$. Figure 13 gives the simulated $S$ parameters of the arced LWA with non-uniform slot distribution and different $N_g$, in which it can be seen that the impedance characteristics of the antenna performed well when $N_g$ was set to 7. Furthermore, the radiation efficiencies of the LWA in the three cases were 38.83%, 44.83%, and 36.86%, respectively. Taking the performance of both the impedance characteristics and radiation efficiency into consideration, $N_g$ was determined to be 7.

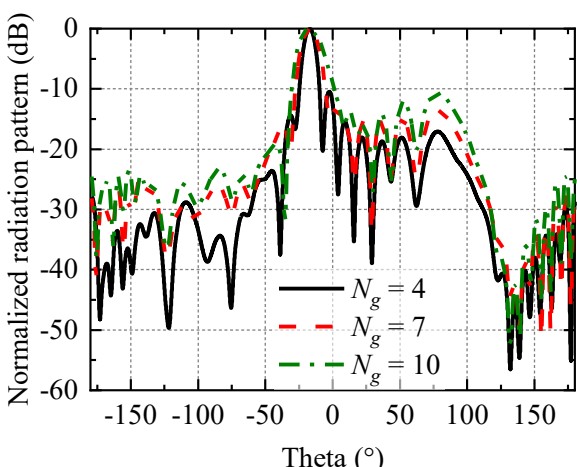

**Figure 12.** Normalized radiation patterns of the proposed arced LWA with different values of $N_g$.

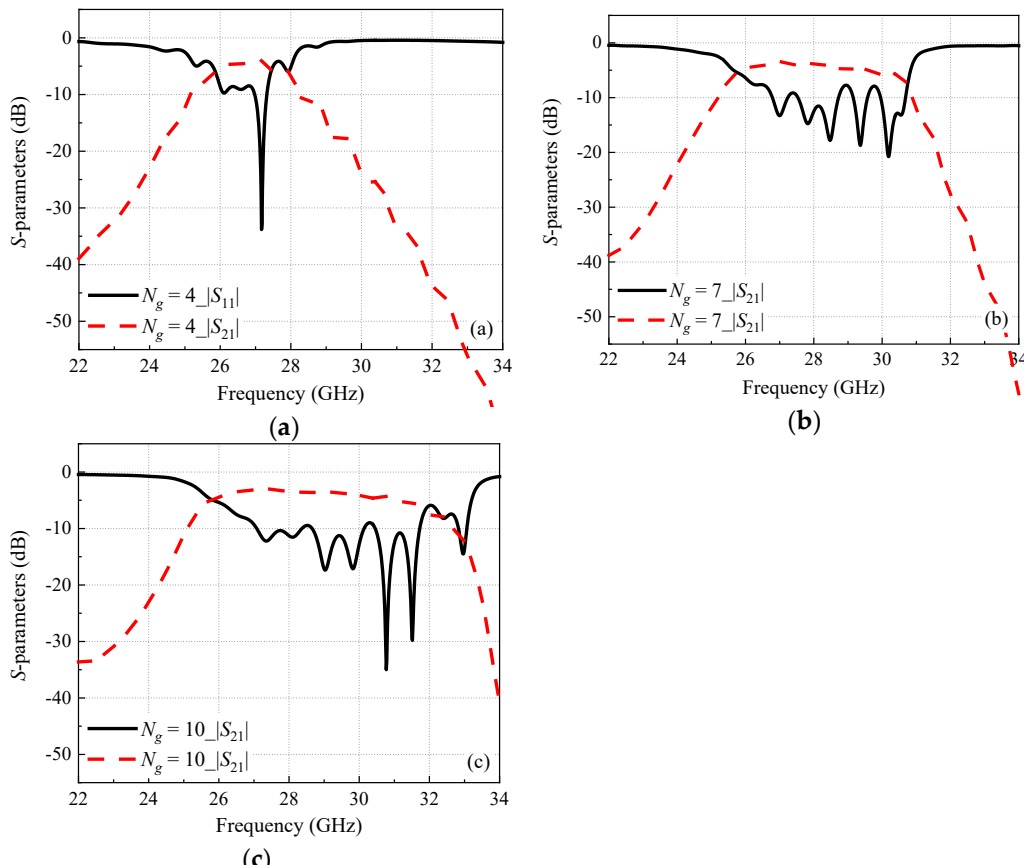

**Figure 13.** Simulated *S* parameters of the proposed arced LWA with different values of $N_g$, (**a**) $N_g = 4$, (**b**) $N_g = 7$, (**c**) $N_g = 10$.

With regard to the feeding structure, a tapered microstrip transmission line is adopted to feed the arced conformal SIW LWA, for which the structural expansion graph is shown in Figure 14 and the simulation structure is shown in Figure 15. Furthermore, Table 4 gives the dimensions of the arced antenna fed by the tapered microstrip line. The ending width of the microstrip line and the thickness of the dielectric layer meet the impedance matching of 50 Ω.

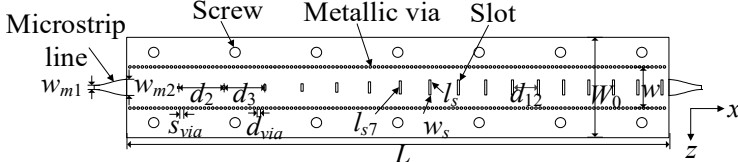

**Figure 14.** Expansion graph of the arced antenna fed by the tapered microstrip line.

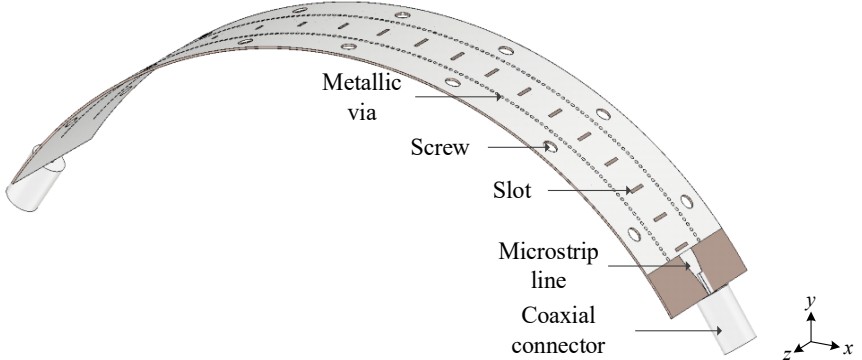

**Figure 15.** Diagram of the arced antenna fed by the tapered microstrip line.

**Table 4.** Dimensions of the arced antenna fed by the tapered microstrip line.

| Parameters | $R$ | $\alpha$ | $w$ | $h$ | $l_s$ | $l_{s1}$ |
|---|---|---|---|---|---|---|
| Value (mm) | 60 | 100° | 8.2 | 0.254 | 3 | 0.375 |
| Parameters | $l_{s2}$ | $l_{s3}$ | $l_{s4}$ | $l_{s5}$ | $l_{s6}$ | $l_{s7}$ |
| Value (mm) | 0.75 | 1.125 | 1.5 | 1.875 | 2.25 | 2.625 |
| Parameters | $w_s$ | $d_{via}$ | $s_{via}$ | $w_{m1}$ | $w_{m2}$ | |
| Value (mm) | 0.4 | 0.5 | 0.8 | 0.8 | 3.2 | |

Figure 16 illustrates the *S* parameters of the arced LWA fed by the tapered microstrip line. It can be seen that the impedance performance was good at 28 GHz. Figure 17 displays the radiation patterns fed by a perfect waveguide port and by the tapered microstrip line. It is clear that the beam direction remained the same for both of the two feedings, which indicates that the arced LWA in this feeding structure can achieve good impedance and radiation characteristics. Furthermore, radiation efficiency of around 45.7% can be achieved by the tapered microstrip-line-fed LWA.

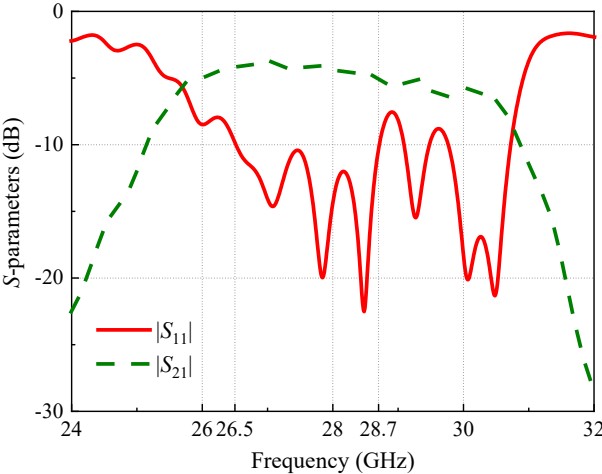

**Figure 16.** *S* parameters of the arced antenna fed by the tapered microstrip line.

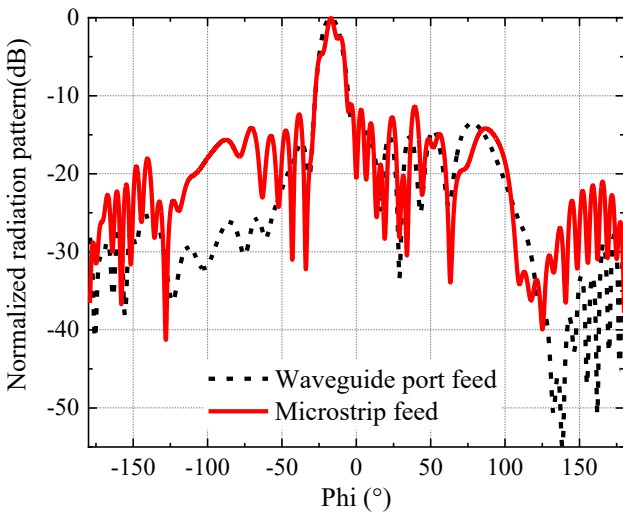

**Figure 17.** Radiation patterns of the arced LWA fed by a perfect waveguide port and by the tapered microstrip line.

## 5. Experimental Verification

In order to verify the proposed arced SIW LWA with non-uniform slot distribution, the LWA was implemented based on PCB processing. A metallic layer of thickness of 0.018 mm was used to achieve better conformal and flexible properties, and gold sinking process was carried out for the metallic layer to avoid oxidation. For convenience of testing, the LWA was conformally fabricated on a nylon support material with an outer radius of R, which was produced by 3D-printing technology. A photograph of the fabricated LWA with the support material is shown in Figure 18. A 2.92 mm coaxial connector was utilized with a probe diameter of 0.64 mm, probe length of 3mm, and outer conductor length of 7.6 mm. The coaxial connector was connected to the microstrip line by welding.

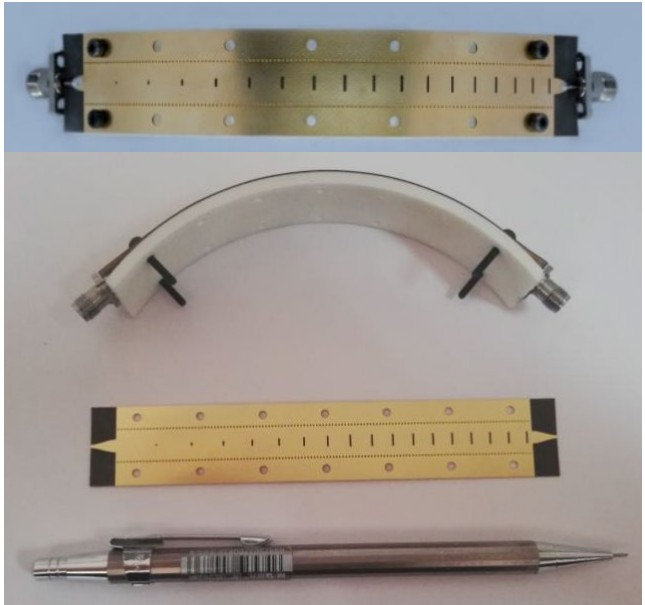

**Figure 18.** The prototype of the arced LWA.

The antenna was tested by utilizing a vector network analyzer in a microwave anechoic chamber. Figure 19 lists the comparison of *S* parameters between simulation and measurement results, in which it can be seen that the $|S_{11}|$s are in good agreement. The

tested $|S_{21}|$ parameters are generally lower than simulated $|S_{21}|$s, which is mainly caused by the loss of coaxial feeding and energy reflection from welding errors.

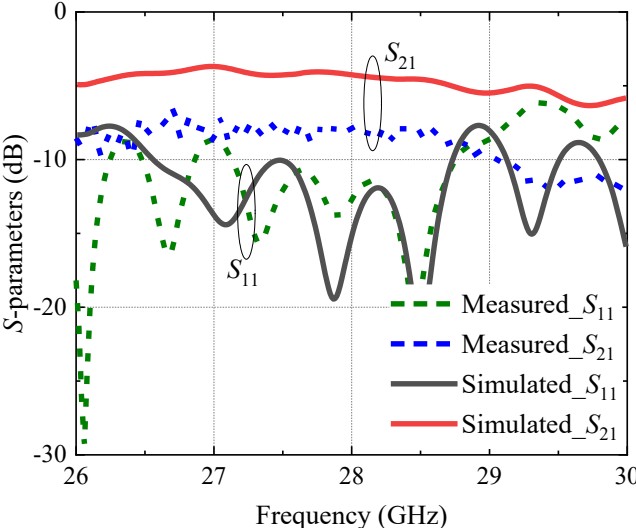

**Figure 19.** Comparison of *S* parameters between simulation and measurement results.

During measurement, the two ports of the antenna under testing (AUT) were connected to a matching load, and a 28 GHz horn antenna was employed as the standard gain antenna, with the aperture center of the E plane aligned with the aperture center of the AUT. The distance between the horn antenna surface and the AUT surface meets the far-field requirement. Figure 20 displays the measurement environment; figure (a) presents the alignment process with the AUT and figure (b) shows the horn antenna employed. By comparison with the standard Ka-band horn antenna, the measured gain of 9.96 dBi was obtained. Figure 21 is the comparison of normalized patterns between simulation and measurement, which indicates that the main beam directions of the two are both basically consistent, as well as the two pattern curves. The measured and simulated main lobe directions are $-16.5°$ and $-18°$, respectively. Consequently, the measurement results verify that the proposed arced LWA with non-uniform slots achieved directional beamforming. Table 5 shows the parameters and performance of several existing conformal antenna. It can be observed that the proposed antenna in this work has a low profile of $0.0017 \lambda_0$ and a small cambered surface area of $1.89 \lambda_0^2$. This is attributed to the compact feed structure, i.e., a leaky wave structure. The slots on the upper side of the SIW are conveniently fed by the structure. A directional beam with 9.96 dBi was generated without any isolated feed network. The beam direction can be rigged by changing the structural parameters before manufacture with the guidance of the established model.

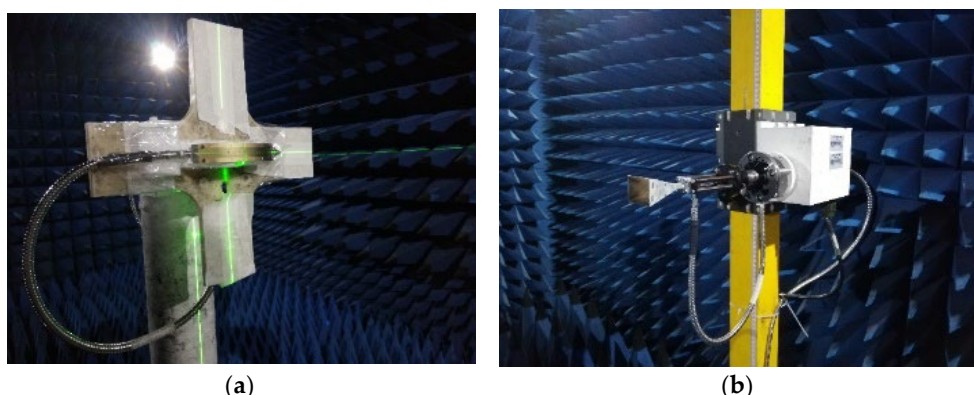

**Figure 20.** Measurement environment: (**a**) antenna under testing, (**b**) horn antenna.

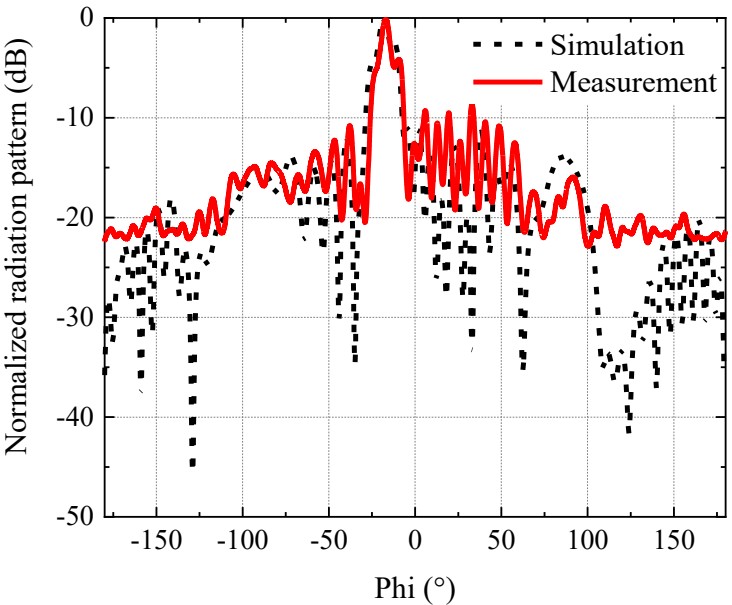

**Figure 21.** Comparison of normalized pattern between simulation and measurement results.

**Table 5.** Performance comparisons.

| Ref. | Type of Antenna | $f_0$ (GHz) | Bandwidth (%) | Profile Height ($\lambda_0$) | Cambered Surface Area ($\lambda_0^2$) | Peak Realized Gain (dBi) |
|---|---|---|---|---|---|---|
| [5] | phased array | 5.5 | 11 | n.a. | n.a. | 8.93 |
| [7] | phased array | 1.58 | 5 | 0.05 | 8.42 | 20.2 |
| [8] | Transmitarray | 10 | 9 | 0.017 | 75.87 | 18.29 |
| [12] | phased array | 2 | 148 | n.a. | 1.15 | n.a. |
| This work | leaky wave | 28 | 5.5 | 0.0017 | 1.89 | 9.96 |

n.a. means the values are not stated in the references.

## 6. Conclusions

In this paper, a low-profile directional conformal LWA based on SIW is proposed. It adopts a microstrip feeding structure with SIW as the transmission line to feed the non-uniform slot array placed on its surface, thereby achieving a specific directional beam. For guiding the design, a radiation-field theoretical model for the arced LWA was built by combining traveling wave theory and circular array theory. By using this model, the radiation characteristics of the arced SIW LWA array were analyzed and the element position distribution of the directional beam with a specific direction was determined. Furthermore, in order to realize the LWA, the traveling wave propagation property of the arced SIW transmission line was studied, showing that the design method of the straight-line SIW can be applied to design the arced SIW. The arced SIW LWA was realized with parameters such as slot distribution, guided wavelength, etc., generated by the model. Finally, the effectiveness of the antenna was verified through actual fabricating and testing, and meanwhile it was confirmed that the proposed theoretical model is able to guide the design of the antenna. The fabricated antenna designed by using the established model achieved a gain of 9.96 dBi with a cambered surface area of 1.89 $\lambda_0^2$. This work provides an alternative solution for conformal antennas for wireless communication with arc-shaped carriers.

**Author Contributions:** Methodology, Y.M., J.W. and F.W.; Validation, Y.M., X.S. and F.S.; Formal analysis, Y.Z. and F.W.; Investigation, X.S., Y.Z. and F.S.; Writing—original draft, Y.M.; Writing—review & editing, Y.M.; Visualization, X.S. All authors have read and agreed to the published version of the manuscript.

**Funding:** This work was supported by the National Nature Science Foundation of China (NSFC) Project under grant no. 62031004.

**Data Availability Statement:** Not applicable.

**Conflicts of Interest:** The authors declare that there are no conflicts of interests regarding the publication of this paper.

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
