# Peer review of "Millimeter-Wave Conformal Directional Leaky-Wave Antenna Based on Substrate-Integrated Waveguide"

_electronics, doi:10.3390/electronics12143111_

Round 1

Reviewer 1 Report

1 The English of the manuscript needs to be improved.

Author Response

Dear Editors,

Thank you and the reviewers very much for your comments and suggestions of our manuscript. We have carefully revised the manuscript according to your comments and suggestions. We would like to express our sincere thanks to you and the reviewers for the constructive and positive comments.

Point to point responses to the reviewer’s comments are listed in following.

We look forward to hearing from you soon. Thanks!

With best regards,

Yours sincerely,

Yuchen Ma

Replies to Editor-in-Chief

The paper has been extended by adding some content. The number of words exceeds 4000 now.

Thank for the suggestion!

Replies to Reviewer 1:

Thank you for your constructive suggestions. Thanks!

This article presents a design of MMW conformal directional LWA based on SIW. The
obtained results are interesting, and before considering the manuscript for further processing, a
few comments are needed:

Comment 1: The English of the manuscript needs to be improved.

Answer: The language has been carefully proofed and corrected.

Thanks again!

Comment 2: The introduction section is too short. Add two or more paragraphs to it, and summarize it in detail.

Answer: The introduction is polished and modified. Several paragraphs has been added, and the methods to designed conformal antennas are summarized.

Thanks again!

Comment 3: The new contributions brought by this research work should be highlighted in the Introduction section. It is not clear what is new in the paper.

Answer: The contributions has been added and highlighted in the last paragraph of Introduction Section.

Thanks again!

Comment 4: The citation method in several places must be changed according to the MDPI format (lines 28, 63, 65, and 66).

Answer: The format of the citation has been corrected according to the MDPI format.

Thanks again!

Comment 5: The novelty, contribution, and main results must be highlighted in the abstract and conclusions sections also.

Answer: The novelty, contribution, and main results are added and highlighted in the abstract and conclusion.

Thanks again!

Comment 6: The title and conclusion mention conformal directional leaky-wave antenna. It would be good to compare results, bandwidth, and dimensions with already published research work.

Answer: The comment is very reasonable and necessary. A comparison and discussion of the exist work and the proposed work has been done in table 5.

Thanks again!

Comment 7: Which was the simulation tool used to analyze the proposed antenna? It'd be nice to tell the readers about the authors' simulation tool and how the simulation has been conducted.

Answer: The simulation is carried out by a commercial full-wave software CST. The simulation can be achieved by: 1. Build the antenna structure, 2. Set the solver of calculation like time domain or frequency domain solver, 3. Set the frequency range and the results interested, 4. Implement the simulation and optimize the performance by analysing the simulation results.

Thanks again!

Reviewer 2 Report

Authors have presented manuscript titled “Millimeter-Wave Conformal Directional Leaky-Wave Antenna based on Substrate Integrated Waveguide”. Work is good and is support with simulated and measured results and discussion. Following comments will be helpful to improve the manuscript.

In abstract, authors should highlight key finding and should support them with numerical values where possible to enhance the significance of presented work.

Introduction is limited, authors should expand the introduction section to provide more insight to the readers on current deployments.

Include references for equations where applicable.

Overall, figures quality and size is not good. Text is not readable being too small. Consider increasing the size of figures and text size as well if required.

Add a comparison table to compare performance of presented design to existing work.

Overall, the work is good and is useful.

Check manuscript for typos and grammatical errors.

Author Response

Dear Editors,

Thank you and the reviewers very much for your comments and suggestions of our manuscript. We have carefully revised the manuscript according to your comments and suggestions. We would like to express our sincere thanks to you and the reviewers for the constructive and positive comments.

Point to point responses to the reviewer’s comments are listed in following.

We look forward to hearing from you soon. Thanks!

With best regards,

Yours sincerely,

Yuchen Ma

Replies to Reviewer 2

Thank you for your constructive suggestions. Thanks!

Authors have presented manuscript titled “Millimeter-Wave Conformal Directional Leaky-Wave Antenna based on Substrate Integrated Waveguide”. Work is good and is support with simulated and measured results and discussion. Following comments will be helpful to improve the manuscript. 

Comment 1: In abstract, authors should highlight key finding and should support them with numerical values where possible to enhance the significance of presented work.

Answer: The key finding is added and highlighted, numerical values such as the gain of the proposed array are added for supporting the key point.

Thanks again!

Comment 2: Introduction is limited, authors should expand the introduction section to provide more insight to the readers on current deployments. 

Answer:  The introduction is extended. More corresponding work is investigated and introduced.

Thanks again!

[17] Wu, Y. F., and Cheng, Y. J. "Proactive conformal antenna array for near-field beam focusing and steering based on curved substrate integrated waveguide." IEEE Transactions on Antennas and Propagation 67.4 (2019): 2354-2363.

[20] Li, M., Liu, Y., Chen, S. L., Hu, J. and Guo, Y. J. "Synthesizing shaped-beam cylindrical conformal array considering mutual coupling using refined rotation/phase optimization." IEEE Transactions on Antennas and Propagation 70.11 (2022): 10543-10553.

[21] Zhang, N., Xue, Z., Zheng, P., Gao, L., and Liu, J. "Synthesis of Low Sidelobe Pattern with Enhanced Axial Radiation for Sparse Conformal Arrays Based on MCDE Algorithm." Electronics 11.22 (2022): 3679.

Comment 3: Include references for equations where applicable. Answer: References for equation (15), (17), (18), (19) are included.

Thanks again!

Comment 4: Overall, figures quality and size is not good. Text is not readable being too small. Consider increasing the size of figures and text size as well if required. 

Answer:  The figures and text in the manuscript are appropriately scaled up.

Thanks again!

Comment 5: Add a comparison table to compare performance of presented design to existing work. 

Answer: Thanks for the constructive suggestion! The comparison table is added as Table 5.

Thanks again!

Comment 6: Check manuscript for typos and grammatical errors.

Answer: The manuscript has been carefully proofed and corrected.

Thanks again!

Reviewer 3 Report

Overall paper looks good. 

1. Include a comparison table

2. English construction needs little improvement

English construction needs little improvement

Author Response

Dear Editors,

Thank you and the reviewers very much for your comments and suggestions of our manuscript. We have carefully revised the manuscript according to your comments and suggestions. We would like to express our sincere thanks to you and the reviewers for the constructive and positive comments.

Point to point responses to the reviewer’s comments are listed in following.

We look forward to hearing from you soon. Thanks!

With best regards,

Yours sincerely,

Yuchen Ma

Replies to Reviewer 3

Overall paper looks good.

Answer: Thanks for the reviewer’s approval to our work!  

Thanks again!

Comment1. Include a comparison table

Answer:Thanks for the constructive suggestion! The comparison table is added as Table 5.

  1. English construction needs little improvement

Answer: The language has been carefully proofed and corrected.

Thanks again!

Round 2

Reviewer 2 Report

Authors have addressed reviewer's comments and manuscript has been improved.